# A Comparison of PKD2L1-Expressing Cerebrospinal Fluid Contacting Neurons in Spinal Cords of Rodents, Carnivores, and Primates

**DOI:** 10.3390/ijms241713582

**Published:** 2023-09-01

**Authors:** Xiaohe Liu, Karen Rich, Sohail M. Nasseri, Guifa Li, Simone Hjæresen, Bente Finsen, Hansjörg Scherberger, Åsa Svenningsen, Mengliang Zhang

**Affiliations:** 1Department of Molecular Medicine, University of Southern Denmark, DK-5000 Odense, Denmark; xliu@health.sdu.dk (X.L.); krich@health.sdu.dk (K.R.); sohail.nasseri@gmail.com (S.M.N.); guifali@health.sdu.dk (G.L.); shjaresen@health.sdu.dk (S.H.); bfinsen@health.sdu.dk (B.F.); aasvenningsen@health.sdu.dk (Å.S.); 2Deutsches Primantenzentrum, GmbH, 37077 Göttingen, Germany; hscherberger@dpz.eu; 3Department of Biology and Psychology, University of Göttingen, 37077 Göttingen, Germany; 4BRIDGE, University of Southern Denmark, DK-5000 Odense, Denmark

**Keywords:** mammalians, spinal cord, central canal, CSF, CSF-cNs, intrinsic sensory neurons, transient receptor potential channels

## Abstract

Cerebrospinal fluid contacting neurons (CSF-cNs) are a specific type of neurons located around the ventricles in the brain and the central canal in the spinal cord and have been demonstrated to be intrinsic sensory neurons in the central nervous system. One of the important channels responsible for the sensory function is the polycystic kidney disease 2-like 1 (PKD2L1) channel. Most of the studies concerning the distribution and function of the PKD2L1-expressing CSF-cNs in the spinal cord have previously been performed in non-mammalian vertebrates. In the present study immunohistochemistry was performed to determine the distribution of PKD2L1-immunoreactive (IR) CSF-cNs in the spinal cords of four mammalian species: mouse, rat, cat, and macaque monkey. Here, we found that PKD2L1-expressing CSF-cNs were present at all levels of the spinal cord in these animal species. Although the distribution pattern was similar across these species, differences existed. Mice and rats presented a clear PKD2L1-IR cell body labeling, whereas in cats and macaques the PKD2L1-IR cell bodies were more weakly labeled. Ectopic PKD2L1-IR neurons away from the ependymal layer were observed in all the animal species although the abundance and the detailed locations varied. The apical dendritic protrusions with ciliated fibers were clearly seen in the lumen of the central canal in all the animal species, but the sizes of protrusion bulbs were different among the species. PKD2L1-IR cell bodies/dendrites were co-expressed with doublecortin, MAP2 (microtubule-associated protein 2), and aromatic L-amino acid decarboxylase, but not with NeuN (neuronal nuclear protein), indicating their immature properties and ability to synthesize monoamine transmitters. In addition, in situ hybridization performed in rats revealed PKD2L1 mRNA expression in the cells around the central canal. Our results indicate that the intrinsic sensory neurons are conserved across non-mammalian and mammalian vertebrates. The similar morphology of the dendritic bulbs with ciliated fibers (probably representing stereocilia and kinocilia) protruding into the central canal across different animal species supports the notion that PKD2L1 is a chemo- and mechanical sensory channel that responds to mechanical stimulations and maintains homeostasis of the spinal cord. However, the differences of PKD2L1 distribution and expression between the species suggest that PKD2L1-expressing neurons may receive and process sensory signals differently in different animal species.

## 1. Introduction

The existence of a system of intrinsic sensory neurons, known as cerebrospinal fluid contacting neurons (CSF-cNs), located in the ependymal layer or the subependymal region around the brain ventricles and the central canal of the spinal cord, has long been demonstrated in different animal species [1,2,3,4,5,6,7,8]. Accumulating evidence indicates that CSF-cNs have the ability to detect environmental changes such as pH, pressure, and flow of CSF in the central nervous system (CNS), and regulation of locomotion [9,10,11,12,13]. A CSF-cN contains a dendritic bulb (or bud) protruding towards the central canal that is supplied with a number of ciliated terminals (stereocilia and kinocilia) to detect chemo- and mechanical stimulation [2,4]. CSF-cNs have been found in several regions in the CNS, such as the hypothalamus [2,14,15,16], the brain stem [5,6,17], and the spinal cord [7,11,18,19]. The majority of the studies concerning CSF-cNs were performed in lower vertebrates including but not limited to turtles [20], frogs [3], zebrafish [11,12], and lamprey [9,10,16], although the studies investigating the distribution, neurochemical properties, and/or functions of CSF-cNs in mammals have been increasing [1,7,19,21,22,23,24]. CSF-cNs exert different biochemical properties identified by many different specific markers including doublecortin (DCX), polysialylated neural cell adhesion molecule (PSA-NCAM), HuC/D (a pan-neuronal marker), gamma-aminobutyric acid (GABA), acid-sensing channels, and polycystic kidney disease 2-like 1 (PKD2L1) channel [5,7,9,17,18,19,25].

PKD2L1, also known as transient receptor potential polycystic 2, is a member of the polycystin protein family, and functions as a calcium-regulated nonselective cation channel. The channel, which was initially identified in the kidney, retina, and heart, was found to be expressed in CSF-cNs in the mouse brain stem and the spinal cord [5,6,17,26]. Later, this channel was also found to be expressed in the CSF-cNs in zebrafish [7,11,12], lamprey [10,16], and some other mammalian species including rat, rabbit, and monkey [7,19]. Since PKD2L1 is a chemo- and mechanical sensory channel that responds to pH and osmolarity alterations and mechanical stimulations its activation may underlie major functional changes in CSF-cNs [5,11,27,28].

Although the PKD2L1-expressing CSF-cNs have been detected in several species including non-mammalian and mammalian vertebrates, a majority of the studies were performed on zebrafish and lamprey, particularly in relation to the cell functions [7,9,11,12,16,28,29]. In mammals, mice were most often used to study the morphology, connectivity, and functional properties of the cells probably due to the availability of PKD2L1 transgenic models [5,6,19,21,22,23,24]. However, the functions of the CSF-cNs in mammals at a systemic level are far from being understood. Because of the differences in anatomical organization between the bodies of fish and mammals, the way CSF-cNs collect mechanosensory information would likely differ [30]. Consequently, the sensory signal processing in the CNS and the final effects on the behavioral output could be different relating to a specific type of neurons. To further understand the functions of PKD2L1 neurons in mammals, it is necessary to have better knowledge of their distribution, morphology as well as their synaptic connections in different mammalian species. One study by Djenoune et al. has shown that CSF-cNs are conserved in the spinal cord in mice and macaque monkeys [7]. However, in that study only selected segments of the macaque monkey spinal cord were investigated, and the focus was on the inhibitory property of the PKD2L1 neurons. In rats, although the existence of the CSF-cNs in the spinal cord is well-documented [18,31], very few data on the distribution of PKD2L1-expressing CSF-cNs exist [19]. To the best of our knowledge, the existence of CSF-cNs in cats has only been studied in the third ventricle using conventional microscopy [32] and in the spinal cord with immunohistochemistry with a vasoactive intestinal polypeptide (VIP) antibody [33]. Whether these neurons expressed PKD2L1 is unknown. There is no data to show the existence of CSF-cNs in the spinal cords of other carnivores.

It is therefore necessary to study the distribution of CSF-cNs systemically in different mammalian species, specifically considering that rats are the common laboratory models and monkeys have a close genetic relation with humans. The main objective of the present study was therefore to map the distribution of PKD2L1-expressing CSF-cNs in the spinal cords of mice, rats, cats, and macaque monkeys using immunohistochemistry and in situ hybridization (only in rats). In addition, we have conducted double-fluorescent immunohistochemistry in rat spinal tissue to characterize the chemical property of PKD2L1-expressing neurons to examine the maturity of these neurons, the cellular location of PKD2L1 channels, and whether these neurons possess the ability to synthesize monoamine neurotransmitters. We chose to use C57BL/6J mice as a control for other species as this mouse strain has been extensively used to study distribution and morphology of CSF-cNs in the spinal cord [5,6,21,22]. Here we show that the distribution pattern of the CSF-cNs was generally similar across these animal species although noticeable differences existed. Thus, our results indicate that CSF-cNs are conserved throughout non-mammalian and mammalian vertebrates and that the cells are not only important in lower-level vertebrates but may also play an essential role in higher levels of vertebrates and probably in humans.

## 2. Results

### 2.1. PKD2L1 Immunoreactivity in C57BL/6J Mouse Spinal Cord

As previously reported by other groups using the same mouse strain (e.g., [6,17,21,26]), PKD2L1-immunoreactive (IR) neurons were distributed throughout the spinal cord from the cervical to caudal level around the central canal (Figure 1). Although we did not perform double immunolabeling with an ependymal cell marker, it was clear that most of the PKD2L1-IR neurons were in the subependymal layer with a few positioned in the ependymal layer (Figure 1A,a,C–F). The cell bodies were usually round or oval, seen in the horizontal or transverse plane, with a diameter of ~12 µm. In most cells a protrusion with a bulbous bud projecting to the central canal could be seen, both in the horizontal and transverse sections. PKD2L1-IR neurons were observed on all sides around the central canal in different spinal segments although in some sections more cells could be seen in the dorsal and ventral sides of the canal (Figure 1C–F). In accordance with previous studies [6,21], a number of PKD2L1-IR neurons were situated away from the central canal, and the presumable dendritic bulbs could be discerned adjacent to the nearby cell bodies rather than projecting towards the central canal (Figure 1B,b). Most of these ectopic PKD2L1-IR neurons were in the ventral side of the central canal usually 100–200 µm from the central canal, either in the gray matter or the white matter (Figure 1B–E). Occasionally PKD2L1-IR neurons were observed in the white matter along the two banks of the ventral median fissure, close to its roof (not shown). However, in other areas, such as the dorsal horn, intermediate region, and ventral horn, a small number of PKD2L1-IR neurons were observed, and the distance of these cells to the central canal could reach as much as 500 µm (Figure 1H–K). In some transverse sections, thin fibers (possibly axons) originating from the PKD2L1-IR cell bodies could be observed running towards the ventral median fissure (Figure 1C,D). These axons might form fiber bundles running longitudinally along both sides of the ventral median fissure (Figure 1G).

### 2.2. PKD2L1 Immunoreactivity in Sprague–Dawley Rat Spinal Cord

The labeling pattern of PKD2L1-IR neurons in the Sprague–Dawley rat spinal cord was similar to that of C57BL/6J mice, i.e., the PKD2L1-IR neurons were distributed throughout the spinal cord from the cervical to caudal level around the central canal, and their dendritic bulbs protruded into the central canal (Figure 2). One difference was that the ectopic PKD2L1-IR neurons were predominantly located on the ventral side of the central canal, and the number of cells seemed lower than those in the mice (Figure 2B,b). Usually, the cells did not reach the ventral median fissure, although occasional cells could be found to do so. Interestingly, there was an area in the sacral segment containing some ectopic PKD2L1-IR neurons, i.e., in the sacral dorsal commissural nucleus located dorsal to the central canal (Figure 2F). Usually, these cells were slightly larger than the PKD2L1-IR neurons around the central canal and their cell bodies were multipolar (Figure 2F,f). In some transverse sections, fibers originating from the ventral side of the central canal could be seen running towards the ventral median fissure (Figure 2C, indicated by arrows). As in mice, these axons might form fiber bundles running longitudinally along both sides of the ventral median fissure (Figure 2C).

### 2.3. PKD2L1 Immunoreactivity in Domestic Cat Spinal Cord

In domestic cats, the PKD2L1-IR neurons were distributed throughout the entire spinal cord from the cervical to caudal level around the central canal, demonstrated by either the existence of the PKD2L1-IR dendritic bulbs or the cell bodies and the bulbs (Figure 3). In comparison with mice and rats, the PKD2L1-IR cell bodies in cats were weakly labeled (Figure 3A,a), and in some transverse sections, the cell bodies were even nonvisible (Figure 3C–E). Even so, the PKD2L1-IR dendritic bulbs protruding to the central canal were densely labeled regardless of whether their connected cell bodies were visible or not (Figure 3A–E). There were three different labeling patterns in relation to these dendritic bulbs when comparing cats with mice and rats. Firstly, most of the bulbs were larger in size compared to those in mice and rats (see Section 2.6 for quantitative analysis). Secondly, in the sections where the cell bodies and the protrusion bulbs were both visible, the number of bulbs was larger than the number of cell bodies (Figure 3a). This may indicate that some of the cell bodies send out more than one dendritic protrusion as is also shown in Figure 3a, where a PKD2L1-IR cell body (the red arrowhead) gave off a stem branch that further divided into three secondary branches, and each of them connected to a bulb (red arrows). Thirdly, the bulbs in the central canal were embedded in a layer of membrane that gave support to the bulbs, so that they did not protrude directly to the lumen. The nature of this membrane is not known and is beyond the scope of the present study. The reason to have such a layer of supporting material may be due to the lumen of the central canal in cats being quite large (can reach to 200 µm in diameter, see Figure 3E) and could help the relatively fixed bulbs to resist the impact of the CSF flow. There seemed to be a thin layer of such membrane, lining the inner surface of the central canal lumen in mice and rats as well, although not as apparent as in cats and monkeys (cf. Figure 1a and Figure 2a vs. Figure 3a,C–F. For monkeys, see next section). As in mice and rats, thin PKD2L1-IR nerve fibers could be seen running from the central canal ventrally, which might finally form fiber bundles projecting rostrocaudally (Figure 3C). Ectopic PKD2L1-IR cell bodies could be detected only in the ventral side of the central canal but not in other regions (Figure 3B,b).

### 2.4. PKD2L1 Immunoreactivity in Macaque Monkey Spinal Cord

Similar to mice, rats, and cats, the PKD2L1-IR neurons were distributed throughout the entire spinal cord from the cervical to caudal level around the central canal (Figure 4). The distribution of the PKD2L1-IR cell bodies and the dendritic bulbs in the central canal was similar to that in the cats, i.e., the PKD2L1-IR cell bodies were weakly labeled (Figure 4A,a), and in some transverse sections the cell bodies were even nonvisible (Figure 4D,G), while the PKD2L1-IR dendritic bulbs in the central canal were densely labeled (Figure 4A–H). The sizes of the bulbs seemed relatively large although some smaller ones were also observed (cf. the bulbs in Figure 1, Figure 2, Figure 3a and Figure 4a) (see Section 2.6 for quantitative analysis). From the sections where the cell bodies and the protrusion bulbs were both well labeled, it seemed that a cell body connected only a single bulb (Figure 4a). As in cats, the bulbs in the central canal were seen to be embedded in a layer of membrane which gave support to the bulbs. A difference between monkeys and mice, rats, and cats was that no PKD2L1-IR fibers were observed running from the central canal ventrally towards the ventral median fissure (Figure 4C,F). Instead, PKD2L1-IR fiber bundles projected rostrocaudally around the central canal on all sides (Figure 4B–H). A small number of PKD2L1-IR cell bodies could be detected a little away from the central canal (~50 µm, red arrowheads in Figure 4F,H). Occasional cell bodies were also observed in the dorsal horn and the intermediate region further away from the central canal (~300 µm, red arrowheads in Figure 4I,J, and the insets).

### 2.5. Quantitative Analyses of PKD2L1-IR Neurons Represented by the Dendritic Bulbs in the Central Canal

To examine whether PKD2L1-IR neurons distributed differentially in different segments of the spinal cord we made a quantitative analysis of the PKD2L1-IR cell numbers around the central canal in all four animal species. As stated in the Materials and Methods section, the PKD2L1-IR cell bodies were not always clearly labeled, especially in cats and monkeys, so we chose to count the dendritic bulbs protruding to the central canal, which were clearly and consistently labeled in all the animal species studied. One drawback of this method could be the risk of underestimating the cell number if the bulbs were lost during the staining process due to the thin shanks connecting the bulbs.

As shown in Figure 5, in the mice the average number of PKD2L1-IR bulbs per 40 µm transverse section was 8.85 ± 0.62, 8.43 ± 0.29, 8.18 ± 0.27, and 7.53 ± 0.08 (mean ± SD) in cervical, thoracic, lumbar, and sacral segments, respectively; in the rats the corresponding numbers were 8.67 ± 1.35, 8.78 ± 1.17, 10.47 ± 1.25, and 7.47 ± 0.94; in the cats they were 8.03 ± 0.26, 8.13 ± 0.71, 8.30 ± +.73, and 10.07 ± 1.18; and in the monkeys they were 6.55 ± 0.55, 9.32 ± 0.85, 9.00 ± 1.49, and 8.57 ± 1.39.

The average number of PKD2L1-IR bulbs was significantly different between the cervical and the sacral segment (8.67 ± 1.35 vs. 7.47 ± 0.94, *p* = 0.006) as well as between the thoracic and sacral segment (8.43 ± 0.29 vs. 7.47 ± 0.94, *p* = 0.006) in the mice (Figure 5A). The remaining segments did not significantly differ (Figure 5A). Despite small variations the number of PKD2L1-IR bulbs did not significantly differ between segments in the other three species (Figure 5B–D).

The average number of PKD2L1-IR bulbs in the entire length of the spinal cord was 8.24 ± 0.32, 8.88 ± 1.18, 8.63 ± 0.72 and 8.36 ± 1.07 per transverse section in mice, rats, cats, and monkeys, respectively, and as seen in Figure 5E, these numbers did not significantly differ between species.

Since the size of the spinal cord in cats and monkeys was much larger than in mice and rats these data indicate that the relative number of the PKD2L1-IR neurons normalized to unit tissue volume would be highest in the mice and lowest in the monkeys although the absolute numbers might show the opposite. The significance of such a distribution pattern in these animal species will be further considered in the Discussion.

### 2.6. The Morphology and Size of the Dendritic Bulbs in the Different Animal Species

By observing the PKD2L1-IR neurons and their dendritic protrusions towards the central canal we noticed that the morphology and sizes of the dendritic bulbs of the PKD2L1-IR neurons from the different animal species were different. Thus, we decided to further examine whether there were measurable differences in the dendritic bulbs in the central canal between the different animal species. Under higher magnification, we found that the shape of the dendritic bulbs was mostly round although some were oval or irregular (Figure 6A–D). Thinner fiber-like structures could be seen protruding from the bulbs making the surface of the bulb a “brush border” as described by Dale et al. [3]. Among these thin fibers some were shorter (~5 µm), and some were longer (~10 µm) corresponding probably to the stereocilia and kinocilia of the CSF-cNs, respectively [2]. Long cilia were commonly seen in the rats, with some bulbs even having two long cilia (Figure 6B). In the mice, some long cilia were observed as well, although not as many as in the rats (Figure 6A). In the cats, long cilia were not commonly seen but short prick-like short cilia were observed around the bulbs, which made these cilia look like a crown around the bulbs (Figure 6C). In the monkeys, multiple medium-sized cilia were often observed connecting with a bulb. These cilia usually formed a brush-like structure from one side of the bulbs (Figure 6D).

Under the microscope we saw that the sizes of the dendritic bulbs were similar between mice and rats. However, they were somewhat larger in cats and monkeys. To further compare the sizes (represented as diameters) of the dendritic bulbs from the different animal species, we measured around 100 bulbs from 2–3 animals from each animal species (for detailed analysis method, see Materials and Methods). As seen from Figure 6E–I, the diameters of the dendritic bulbs were quite similar in the mice and rats. Thus, in the mice they ranged from 3.0 µm to 5.0 µm (mean ± SD = 3.86 ± 0.41; median = 3.85), and in the rats they ranged from 3.2 µm to 5.0 µm (mean ± SD = 3.82 ± 0.38, median = 3.80); whereas in the cats the range was between 3.8 µm and 7 µm (mean ± SD = 4.94 ± 0.75, median = 4.94), and in the monkeys from 5.2 µm to 8.8 µm (mean ± SD = 6.42 ± 0.90, median = 6.18). Statistical analysis of the average diameters of the dendritic bulbs demonstrated that the size was significantly larger in the monkeys compared to all three other animal species (*p* < 0.0001) (Figure 6J). The size was also significantly larger in cats compared to that of mice and rats (*p* < 0.0001) (Figure 6J). No significant difference was seen between rats and mice (Figure 6J) (*p* > 0.999).

### 2.7. PKD2L1 Were Expressed in Immature Neurons and Co-Expressed with L-Amino Acid Decarboxylase (AADC) in Rats

It has previously been shown that in mice PKD2L1-expressing CSF-cNs are immature neurons [6]. To examine if this is the case in rats as well, we performed double-immunofluorescent labeling with PKD2L1 together with NeuN (neuronal nuclear protein, a mature neuronal marker) or DCX (an immature neuronal marker) antibodies in rat spinal cord sections. Here we found that PKD2L1-IR neurons around the central canal did not express NeuN (Figure 7A1–A3) though they indeed expressed DCX (Figure 7B1–B3). These results are in agreement with previous data suggesting that PKD2L1-IR neurons around the central canal are immature neurons. In addition, by double labeling with MAP2 (microtubule-associated protein 2, a dendritic marker) antibody, we were able to confirm that the protrusions from the PKD2L1-IR neurons were dendrites (Figure 7C1–C3).

In our previous studies, we have found that in the rat spinal cord CSF-cNs expressed AADC—a necessary enzyme for the synthesis of some of the monoamine transmitters, such as serotonin and dopamine [34,35]. To investigate whether the CSF-cNs expressing PKD2L1 and AADC belong to the same population we performed double labeling with PKD2L1 and AADC antibodies. The results showed that all the PKD2L1-IR neurons also expressed AADC, strongly suggesting that they are the same population in the rat spinal cord (Figure 7D1–D3).

### 2.8. PKD2L1 mRNA Expression in the Rat Spinal Cord

Mouse and rat are both commonly used laboratory animal species. PKD2L1 expression has been extensively investigated in the mouse spinal cord, including its distribution at both mRNA and protein levels (e.g., [6,17,21,26]), but data concerning its distribution in rats is lacking [19]. The expression of PKD2L1 mRNA was therefore investigated in the rat spinal cord using in situ hybridization. As shown in Figure 8A from a thoracic section, the hybridization signal was only observed around the central canal, but not in any other areas. The hybridization signal was observed both in the cell bodies and dendritic protrusions towards the central canal (Figure 8a, arrows). This mRNA expression pattern was observed in the investigated spinal segments (cervical, thoracic, and lumbar). Hybridization for “house-keeping gene” glyceraldehyde 3-phosphate dehydrogenase (GAPDH) mRNA showed strong hybridization signals in the gray matter spinal neurons (Figure 8B); while performed with buffer only, or when the sections were treated with RNase A, hybridization showed no signal (Figure 8C). The latter confirmed the specificity of the in situ hybridization reaction.

## 3. Discussion

The present study systematically examined the distribution of PKD2L1-IR neurons in the spinal cords of four different mammalian animal species, which included mice, rats, cats, and macaque monkeys. Among these four animal species, the PKD2L1-expressing neurons in mice have been extensively investigated to date (e.g., [5,6,7,19,21,22,23,24,26]). For rats, although there are some studies that examined the distribution and chemical properties of the CSF-cNs in the spinal cord, very few data exist concerning the distribution of PKD2L1-expressing CSF-cNs neurons [19]. Information concerning the PKD2L1-expressing CSF-cNs in monkeys is even more sparse [7]. To our knowledge, our study is the first to show the expression of PKD2L1 in the spinal cords of cats or any other carnivores although the existence of CSF-cNs has been reported [33]. Therefore, our study extends the knowledge about the distribution of PKD2L1-expressing CSF-cNs in mammals and suggests that the PKD2L1-expressing CSF-cNs may be similarly distributed across vertebrates and invertebrates and may exert similar or different functions in different animal species.

In this study, PKD2L1-IR neurons were found throughout the rostrocaudal extension of the spinal cord across all four animal species, where they were predominantly located around the central canal although varied numbers of the cells were also found in other regions—so-called distal or ectopic PKD2L1 neurons [19,21]. Regardless of the labeling density of the cell bodies the distribution pattern was generally similar among all mammalian species/strains studied so far, which include C57BL/6J and Balb/C mouse, Wistar rat, New Zealand rabbit, and macaque monkey but the C57BL/6N mouse [6,7,19,21]. The distribution of PKD2L1-IR neurons in C57BL/6J mice from our study was consistent with the published data in the same substrain [6,7,19,21]. However, the labeling pattern was different from another mouse substrain—C57BL/6N, where more distally located PKD2L1 neurons were found [19] (see below for further discussion). The distribution of PKD2L1-IR neurons in rats was similar to C57BL/6J mice, i.e., the majority of the cells were located around the central canal with a smaller number of the cells located on the ventral side of the central canal and other regions. The pattern seems similar between the two rat strains studied: Sprague–Dawley rats in this study and Wistar rats reported by Gombalová et al. [19]. The distribution pattern of the PKD2L1-IR neurons in cats and macaque monkeys was similar, i.e., most PKD2L1-IR neurons were found to be located around the central canal and fewer cells in the distal areas. CSF-cNs have been described in cat spinal cord with immunohistochemistry for VIP [33], but there was no study to investigate the expression of PKD2L1 in cat spinal cord before. Although the existence of CSF-cNs in macaque monkey spinal cord has been suggested using VIP immunohistochemistry by Lamotte [33], PKD2L1-expressing neurons around the central canal in this species were first confirmed by Djenoune et al. [7]. In that study, only qualitative data were described from two monkeys in selected spinal segments. The labeling pattern and the morphology of the PKD2L1-expressing neurons in macaque monkeys from ours and Djenoune et al. [7] were consistent, namely, the PKD2L1-immunoreactive products were rich in the dendritic bulbs towards the central canal but weakly labeled or absent in the cell bodies. Such a weak cell body immunoreactivity was different compared to mice and rats. In the rodents, dense PKD2L1 immunoreactivity was seen both in the cell bodies and dendritic bulbs. The reasons causing such a difference need to be further investigated. The results could be real or due to the preparation of the tissue samples. Firstly, the cats and monkeys had all undergone electrophysiological experiments before the spinal cord tissues were collected. To what extent the electrophysiological experiments (placing the electrodes in the brain and the spinal cord) could have influenced PKD2L1 expression is unknown. To find out such an influence we need cats and monkeys without prior electrophysiological experiments as control subjects. Unfortunately, there are no such experimental animals available now, at least in Denmark. Secondly, the spinal cord tissues from cats and monkeys used in our study were not fresh. They were obtained after physiological experiments which usually lasted a long time (many hours to many days), and since then the tissues have been kept at −80 °C for many years. Such a tissue condition might not give an optimal immunolabeling result or could potentially give varying results. The latter possibility is demonstrated by a relatively better cell body labeling from one cat and one monkey spinal cord. However, it could also be true that more PKD2L1 proteins were in the dendrites than in the cell bodies. Our in situ hybridization results from rats have shown PKD2L1 mRNA expression in the dendritic protrusions towards the central canal (see Figure 8a). Although we did not perform in situ hybridization experiments in the spinal cord from cats and monkeys, the result may indicate that PKD2L1 proteins can be synthesized directly in the dendritic bulbs and do not need to be synthesized in the cell bodies first and transported to the dendrites. Indeed, the labeling pattern of VIP in cat and monkey spinal cords showed by Lamotte [33] displayed a similar labeling pattern with PKD2L1 immunolabeling, with more exhibiting labeled dendritic bulbs in the central canal and weaker cell bodies in the ependymal layer.

The estimation of cell numbers by counting the PKD2L1-IR dendritic bulbs in different parts of the spinal cord revealed a very similar distribution of the PKD2L1-IR neurons for all the four animal species (ca. eight bulbs per 40 µm section) in our study. Gombalová et al. [19] calculated the abundance of the PKD2L1-labeled CSF-cNs in lumbar spinal cord in C57BL/6N, C57BL/6J, and Balb/C mouse, Wistar rat, and New Zealand rabbit, and their results also showed nonsignificant differences among these species (six–nine cells in a 10 µm thick Z-stack). These results suggest that smaller animals possess a relatively higher number of PKD2L1-IR neurons in relation to the tissue volume. We also made a quantification of PKD2L1-IR neurons in different spinal segments for the different animals, and the results indicated an equal distribution of PKD2L1-IR neurons in the different segments for all the animal species except for the C57BL/6J mice where the cell number in the sacral segments was slightly, though statistically significantly, lower than in cervical and thoracic segments. Gombalová et al. [19] studied the distribution of PKD2L1-IR neurons in different segments of the spinal cord in the C57BL/6N mice, showing no significant differences between the spinal segments. These results suggest that the CSF-cNs are related to some functions that involve all parts of the spinal cord, rather than functions restricted to specific segments. We need to point out that the cell numbers based on the dendritic bulbs from 40 µm thick sections in our study were much lower than the cell numbers counted from 10 µm Z-stacks in mice if their numbers were converted to 40 µm Z-stack [19,21]. We believe that this might be due to technical issues. As described in the Results section, our counting method might underestimate the cell numbers due to the loss of the dendritic bulbs in the central canal during tissue processing. However, counting on 10 µm Z-stacks might also overestimate the cell numbers if stereology was not used.

We did not make a quantitative analysis of the distal PKD2L1 neurons in any of the species investigated, although our observation indicated more PKD2L1-IR neurons around the central canal than in the regions distal from it. According to Gombalová et al. [19] about 15%, 23%, 27%, and 7% of all the PKD2L1-IR neurons were distal neurons in the lumbar spinal cord in C57BL/6J mice, Balb/C mice, Wistar rats, and New Zealand rabbits, respectively; whereas in C57BL/6N mice the value was 78%. Another study by Jurčić et al. [21] reported about 28% distal PKD2L1 neurons in C57BL/6J mice on average from the medulla and the cervical plus lumbar segments of the spinal cord. We noticed that the criteria to define the distal PKD2L1 neurons were not the same in the two existing studies. Jurčić et al. [21] counted PKD2L1 neurons 50 µm away from the central canal as distal neurons, whereas Gombalová et al. [19] counted the neurons that were not in or adjacent to the ependymal cells as distal neurons. This means that in Gombalová et al. [19] some neurons might be within 50 µm distance from the central canal and were still counted as distal neurons; thus, the percentage of the distal neurons might become somewhat higher. Paradoxically, when looking at the percentages presented in C57BL/6J mice by Gombalová et al. [19] (15%) and Jurčić et al. [21] (28%), this seemed not to be the case. Thus, the distance of the PKD2L1 neurons to the central canal might not be an important factor in determining their functions. The key issue is instead whether these two populations of PKD2L1 neurons belong to two different morphological and functional categories. From the published data and the results from the present study this seems to be the case. The first reason is that a proximal PKD2L1 neuron around the central canal usually has a dendritic protrusion towards the central canal in all the species studied [6,7,9,11,19,21], though this is not the case for distal PKD2L1 neurons. Concerning the distal PKD2L1 neurons, some have a dendritic protrusion to the central canal, while some have a dendritic protrusion to the anterior median fissure, and the rest have a dendritic ending in the spinal parenchyma ([19,21]; present study). The second reason is that the morphology of a group of distal PKD2L1 neurons is different from the proximal PKD2L1 neurons. For the proximal PKD2L1 neurons and a part of distal PKD2L1 neurons with a dendritic protrusion to the central canal, cell bodies are usually small (~10 µm in diameter) and bipolar. The other group of distal PKD2L1 neurons, especially those that are further away from the central canal, have larger multipolar cell bodies [6,19,21]. As shown in Figure 2F,f, the sizes of the PKD2L1-IR neurons in the sacral dorsal commissural nucleus in the rat spinal cord were clearly larger than the PKD2L1-IR neurons around the central canal (cf. Figure 2a,f), and no dendritic protrusions could be traced from these neurons connecting to the central canal. These findings together suggest that there are two types of PKD2L1-expressing neurons in the spinal cord: one type belongs to the CSF-cNs with their dendritic protrusions project to the central canal or the anterior median fissure (where CSF can also be contacted), and the other type restricts their dendritic branches in the spinal parenchyma, and may receive information from the internal environment of the spinal cord rather than from the CSF. The distal PKD2L1 neurons may be further categorized into several different types according to their position, morphology, connectivity and perhaps function.

The most interesting and striking finding in the present study was that the sizes of dendritic bulbs varied in different animal species. The dendritic bulbs from mouse and rat spinal cord shared the smallest size, the ones from cats were in the middle, and those from the monkeys were the largest. The average diameter of the dendritic bulbs in the monkeys was ~1.7-fold of that in the mice and rats, and there was no overlap in size distribution between mice/rats and monkeys. There seemed to be a tendency towards an increase in size of the dendritic bulbs with the size of the animals. As the larger animals tended to have a lower number of PKD2L1 neurons per unit spinal tissue volume we suggest that the larger dendritic bulbous size may function as a compensation mechanism for the smaller cell number, having a larger number of receptors on the surface of each dendritic bulb. We acknowledge that the processes of freezing, cryoprotection, and subsequent thawing for the cat and monkey spinal cord samples might have influenced the tissue’s volume and morphology. Usually after many years of storage in a freezer, the tissue may shrink to a certain degree. If this is the case, the size of the cell bodies and dendritic bulbs would become smaller than in fresh samples. Since we saw a larger dendritic bulb size in cats and monkeys, if the samples were fresh the size would be even larger, and thus the differences between cats/monkeys and mice/rats would be even more significant. Since most of the dendritic bulbs were round (a putative shape of dendritic bulbs) in the cats and monkeys the influence on the cell morphology might be minimal, if any. Another interesting finding is that we observed numerous cilium-like protrusions from dendritic bulbs in all four animal species investigated (see Figure 6), with most of the cilia being shorter and one or two cilia being longer. It is well known that the dendritic bulbs bear kinocilia and stereocilia, with the number of the first type being frequently just one and the latter type being many [11,36]. However, it has previously been found that in lamprey spinal cord, a dendritic bulb occasionally can have more than one kinocilia [37]. The function of the kinocilia is to detect mechanical stimulation, such as the flow of the CSF, whereas the function of the stereocilia is to detect the chemical stimulation, such as pH in the same medium [2,37]. It is most likely that the shorter cilia that we observed correspond to the stereocilia and the longer ones to kinocilia although their exact identity needs to be confirmed with super resolution microscopy.

It has previously been shown that the axonal fibers from the CSF-cNs around the central canal project longitudinally with longer projection rostrally in all the animal species studied [12,18,21,24,38]. In zebrafish, mice, and rats the axonal fibers run mostly below the central canal and some run even further ventrally reaching the anterior median fissure and then turn and run longitudinally [12,18,24]. Although it is not easy to track single fibers with immunohistochemistry, we could see that in mice, rats, and cats the PKD2L1-IR axonal fibers ran ventrally towards the anterior median fissure, and that the fiber bundles were clearly visible on both sides of the fissure (Figure 1, Figure 2 and Figure 3). However, in monkeys we could not distinguish such fibers and bundles; rather, dense fiber-like structures could be seen on different sides of the central canal—outside the ependymal layer (Figure 4). This difference indicates that in monkeys the axons may turn rostrocaudally immediately after leaving their cell bodies. This may be due to the larger size of the central canal in larger animals so that the axonal fibers find their optimal way to reach their destinations.

Although the main purpose of the investigation was not to characterize the chemical properties of the spinal PKD2L1 neurons, we performed double fluorescent labeling with PKD2L1 and NeuN, DCX, MAP2, or AADC antibodies in rat spinal cord. We did not perform double fluorescent labeling in cat and monkey spinal cord tissue mainly due to the inconsistent staining pattern for PKD2L1-IR cell bodies. The results with PKD2L1 and NeuN or DCX immunostaining confirmed that the PKD2L1-IR CSF-cNs around the central canal were immature neurons, as they were negative for NeuN and positive for DCX. Some previous studies have been able to detect a positive yet low level of NeuN expression in the PKD2L1 neurons in adult mice [6,19,21,26]. The discrepancy between these results might not be due to the differences between the species, but rather caused by the sensitivity of the detecting techniques. Our results showed a pronounced DCX expression, but it was limited to the PKD2L1 cell bodies and dendritic protrusions (Figure 7B2), a result confirming the immature property of the CSF-cNs. Our DCX labeling pattern was similar to that shown in the mouse CSF-cNs although the DCX antibodies were not from the same origin [6,21]. The intense labeling in our image might be due to the imaging processing rather than due to the specificity. The results obtained using PKD2L1 and MAP2 immunolabeling confirmed the dendritic property of the protrusions towards the central canal. All these results agree with previous studies in mice and rats [6,7,18,21,26,31]. We also found that in rats, PKD2L1-IR neurons exclusively express AADC—a necessary enzyme for monoamine neurotransmitter production, including the generation of serotonin, dopamine, and several trace amines [34,35,39]. In a previous study by Orts-Del’immagine et al. [6] it was shown that, in the medulla, PKD2L1 neurons did not express serotonin or tyrosine hydroxylase, suggesting that PKD2L1 neurons are neither serotoninergic nor catecholaminergic. Their co-expression with AADC indicates that CSF-cNs at least in rats—and probably also in other mammals—have the potential to synthesize monoamine neurotransmitters from their precursors [34,35]. In mammals the monoamine neurotransmitters are mainly provided from the brain, thus the spinal cord may gradually lose the ability to directly supply these substances during the evolution [40]. However, in animals at a lower evolutionary level, e.g., chicken, xenopus, zebrafish, and lamprey, the CSF-cNs in the hypothalamus or spinal cord may supply monoamine neurotransmitters such as serotonin and dopamine at least locally [37,40]. In the present study we did not perform a double immunolabeling with the PKD2L1 antibody and an antibody for GABA, largely due to technical reasons. Abundant evidence from other laboratories has shown that most, though not all, of the CSF-cNs are GABAergic in animal species including zebrafish, lamprey, mice, rats, and monkeys [7,18,41,42]. Whether this is the case for cats is unknown and it is worth examining further.

The function of the CSF-cNs has for a long time been a question that has bewildered scientists. Since the cells contain both chemical and mechanical receptors it has been demonstrated that the cells can sense both chemical and mechanical stimulations, such as the pH changes and CSF flow, to help maintaining a homeostasis of the spinal cord [2,4,9]. Recent studies from fish including zebrafish and lamprey have indicated CSF-cNs’ function in postural control and locomotion and the maintenance of spine curvature [11,12,16,28,38]. The research concerning the function of the CSF-cNs in mammals has been sparse, and the roles of CSF-cNs in mammals seem not to be the same as in fish [30]. There are only two studies that investigate the functions of CSF-cNs at a systemic level in mice: one study showed that the activation of the CSF-cNs do not affect general motor activity or the generation of locomotor rhythm and pattern, but is involved in skilled movements [22]. The other study showed that CSF-cNs can affect the moving speed and the step frequency [24]. More work is thus needed to explore the functions of the CSF-cNs in mammals at different levels. To be able to evaluate the function of a neuron, it is necessary to know its morphology, its chemical properties, and its input and output connectivity. As discussed above, there might be several types of PKD2L1 neurons (not least CSF-cNs) in the spinal cord and each of these subtypes may have different functions depending on their anatomical profiles. In fish and mice, the axonal fibers of CSF-cNs were found to terminate in the ventral horn motor neuron region and the regions around the central canal [22,24,38]. There are many kinds of interneurons around the central canal, e.g., premotor neurons, autonomic neurons, and neurons relating to nociception [43,44,45]. It is therefore extremely important to study the anatomical and functional connectivity of all types of PKD2L1 neurons to understand their functions.

In conclusion, in this study we have demonstrated the presence of PKD2L1-IR neurons around the central canal and in other regions in the spinal cord in four different mammalian species including rodents, carnivores, and primates. The distribution of PKD2L1-IR neurons was almost similarly distributed in different parts of the spinal cord in all these species. The relatively larger size of the dendritic bulbs in cats and macaque monkeys compared to mice and rats indicates a larger capacity of each PKD2L1 neuron to receive the incoming stimulation in animals with a larger body size. Thus, the results added further evidence for the conservation of PKD2L1 neurons (including CSF-cNs and PKD2L1 neurons in other areas) across different animal species.

## 4. Materials and Methods

### 4.1. Animals and Tissue Preparation

In this study mice, rats, cats, and macaque monkeys were used as experimental animals. The experimental protocol for mice and rats was approved by the Danish Animal Experimentation Inspectorate (permit no. 2020-15-0201-00566). The cat and macaque monkey spinal cords were obtained from different resources (see below).

*Mice:* Adult male C57BL/6J mice (2–3 months old, n = 8, Taconic, Silkeborg, Denmark) weighing 20–30 g were anaesthetized with 60 mg/kg pentobarbital and then perfused with 4% paraformaldehyde (PFA) in phosphate-buffered saline (PBS). The spinal cord was removed and post fixated in PFA at 4 °C overnight, and then moved to PBS with 30% sucrose. The spinal cord from different segments (cervical, thoracic, lumbar, and sacral) was then cut either transversely or horizontally into 40 µm thick sections on a sliding microtome (Microm HM450, ThermoScientific, Roskilde, Denmark) connected to a freezing unit (Microm KS34, ThermoScientific, Roskilde, Denmark). The spinal sections from the mice were only used for immunohistochemistry.

*Rats:* Adult male Sprague–Dawley rats (4–5 months old, n = 8, Taconic, Silkeborg, Denmark) weighing 300–500 g were used. Six rats were used for immunohistochemistry and two were used for in situ hybridization. The rats used for immunohistochemistry were perfused with 4% PFA, and the spinal cords were processed the same way as the mice. The rats used for in situ hybridization were euthanized in a CO_2_ chamber. When the respiration stopped, CO_2_ flow was stopped, and the animals were checked for life signs after a minimum of 5 min. The spinal cord was dissected and divided into different segments (cervical, thoracic, lumbar, and sacrocaudal), which were then frozen with dry ice and stored at −80 °C. The cervical, thoracic, and lumbar spinal segments were cut transversely into 20 µm thick sections using a Leica CM3050S cryostat (Leica Systems Triolab A/S, Brøndbyl, Denmark), and the sections were mounted on super frost slides and stored at −80 °C until further processing.

*Cats:* Spinal cords from 3 adult male domestic cats (1–6 years old, 2.5–4 kg) were obtained from the University of Copenhagen initially used for electrophysiological experiments between 2008 and 2009 [46]. The experimental protocol was approved by the Danish Animal Experimentation Inspectorate (permit no. 2005/561–966). The cats were perfused immediately with 4% PFA after the termination of the electrophysiological experiments. The spinal cords were then removed and separated into different segments, cryoprotected in PBS with 30% sucrose, and stored at −80 °C. Selected spinal segments were cut into 40 µm thick sections either transversely or horizontally before being processed with immunohistochemistry.

*Macaque monkeys:* Spinal cords from 3 adult macaque monkeys (2 male, 1 female, 10–13 years old, 7.4–8.9 kg) were obtained from the German Primate Center in Göttingen, Germany. The monkeys were killed after long-term electrophysiological and behavioral experiments [47,48] conducted under the authorization of the Animal Welfare Division of the Office for Consumer Protection and Food Safety of the State of Lower Saxony, Germany (permit no. 33.9.42502-04/032/09 and 33.9-42502-04-11/0448). After cardiac perfusion with 4% PFA, the spinal cords were removed and separated into different segments. For technical reasons, the cervical spinal cord from one monkey was not successfully removed, but from two other monkeys the entire spinal cord was obtained. After cryoprotection in PBS with 30% sucrose they were frozen on dry ice and stored at −80 °C. Before immunolabeling, selected spinal segments were cut into 40 µm thick sections either transversely or horizontally. The spinal sections from the macaque monkeys were only used for immunohistochemistry.

### 4.2. Peroxidase Immunohistochemistry

To achieve an overall distribution pattern of PKD2L1-IR CSF-cNs in the spinal cords from different animal species, both transverse and horizontal sections from cervical, thoracic, lumbar, and sacral segments from mice, rats, cats, and monkeys were processed with single peroxidase immunohistochemistry. Sections were first incubated in 0.3% hydrogen peroxide in PBS for 30 min at room temperature to block the intrinsic peroxidase. After multiple washes with PBS + 0.1% triton X-100 (PBST) the sections were preincubated in a mixture of PBST with 5% normal goat serum (NGS) (Biowest, Lakewood Ranch, FL, USA) + 2% bovine serum albumin (BSA) (Sigma-Aldrich, Soeborg, Denmark) for 1 h at room temperature. The sections were then incubated with primary rabbit anti-PKD2L1 antibody (1:2000–4000; #AB9084, Merck Millipore, Burlington, MA, USA) in PBST with 5% NGS + 2% BSA for 2 overnights at 4 °C. After a thorough rinse in PBST, the sections were incubated in biotinylated goat anti-rabbit-IgG secondary antibody (1:500, #E0432, DAKO, Glostrup, Denamrk) in PBST with 2% NGS + 1% BSA for 1 h at room temperature. Subsequently, the sections were washed in PBST and incubated in an avidin-biotin complex (PK-6100, Vector Laboratories, Burlingame, CA, USA) for 30 min. After being washed in PBS and Tris-buffered saline (TBS) the sections were stained in 0.05% 3,3′-diaminobenzidine (DAB) in TBS with 0.01% hydrogen peroxide for 5–15 min. Sections were mounted on slides and dried before being coverslipped with DPX (Merck Millipore, Burlington, MA, USA). The specificity of the rabbit anti-PKD2L1 antibody has been verified in many studies (e.g., [6,7]). A negative control was always performed in this study with the primary antibody omitted for both peroxidase and fluorescent immunohistochemistry. No positive control was performed for any of these primary antibodies.

### 4.3. Fluorescent Immunohistochemistry

Double fluorescent immunohistochemistry was performed only in the sections of rats’ spinal cord. To assess whether PKD2L1 was expressed in mature or immature neurons, double fluorescent labeling was performed using rabbit anti-PKD2L1 (1:1000) and mouse anti-neuronal nuclear protein (NeuN, 1:1000; #Mab377, Merck Millipore, Burlington, MA, USA) or guinea pig anti-double cortin (DCX, 1:50; AB2253, Merck Millipore, Burlington, MA, USA). To confirm that the protrusions of the PKD2L1-IR neurons to the central canal were dendrites, rabbit anti-PKD2L1 (1:1000) and chicken anti-microtubule-associated protein 2 (MAP2, 1:2000; #ab5392, Abcam, Cambridge, UK) were used. To identify whether PKD2L1-IR neurons can synthesize monoamine neurotransmitters, rabbit anti-PKD2L1 (1:1000) and sheep anti-aromatic L-amino acid decarboxylase (AADC, 1:100; #AB119, Merck Millipore, Burlington, MA, USA) were used. The staining protocols for the different double immunostaining were similar. Transverse sections from the cervical, thoracic, and lumbar segments of rats were used. In brief, the sections were incubated first in PBST with 5% NGS or normal donkey serum (NDS) (Biowest, Lakewood Ranch, FL, USA) + 2% BSA for 1 h at room temperature to block the unspecific immunoreactivity, then they were incubated in above primary antibody pairs for 2 overnights at 4 °C. After thorough rinsing with PBST, the sections were incubated in the corresponding secondary antibody pairs in 0.1 M PBST with 2% NGS or NDS for 2 h at room temperature. The secondary antibody pairs included goat anti-rabbit Alexa Flour 594 (#A-11012) + goat anti-mouse Alexa Flour 488 (#A-11001), goat anti-rabbit Alexa Flour 594 + goat anti-guinea pig Alexa Flour 488 (#A-11073), goat anti-rabbit Alexa Flour 488 (#A-11008) + goat anti-chicken Alexa Flour 594 (#A-11042), and donkey anti-rabbit Alexa Flour 594 (#A-21207) + donkey anti-sheep Alexa Flour 488 (#A-11015) with a concentration of 1:200–1500. All the fluorescent secondary antibodies were from Invitrogen/ThermoFisher Scientific (Waltham, MA, USA). Sections were then rinsed in 0.1 M PBS, mounted on slides and coverslipped with fluorescence mounting medium (S3023, Dako, Glostrup, Denamrk).

### 4.4. In situ Hybridization

In situ hybridization was used to detect PKD2L1 mRNA expression. The oligodeoxynucleotide probe was PKD2L1-rattus norwegicus with an alkaline phosphatase (AP) conjugated 5′amine-C6 (5′/AP/CAA GAT GAC CAC CAG GTC CAG AAT GTT CCA 3′) (manufactured by LGC Biosearch Technologies, Lystrup, Denmark).

In situ hybridization was performed under RNase free condition. The 20 µm thick sections from the cervical, thoracic, and lumbar segments were dried at 55 °C in an oven for 10 min, then placed in 96% ethanol in glass jars for 3 h at room temperature. RNase A sections were treated with RNase A for 2 h. All sections were airdried at room temperature for 45 min. Glasses were placed in a hybridization tray, and sections were hybridized in 100 µL hybridization buffer with a probe (7.5 pmol/mL) under coverglass in a 37 °C oven overnight [49]. Sections were then washed three times for 30 min in pre-heated saline-sodium citrate (SSC) buffer in a 55 °C warm oven. Then sections were rinsed in Tris-HCl buffer (pH 9.5) and subsequently developed using a mixture of developing buffer and substrate solution which contained nitro blue tetrazolium chloride (NBT) and 5-Bromo-4-chloro-3-indolyl phosphate (BCIP). Sections were developed for 3 overnights and then washed in 37 °C water for 1 h. Sections were coverslipped with Aquatex (Merck Millipore, Burlington, MA, USA).

To ensure the same high quality of all in situ hybridizations experiments parallel sections were hybridized with a probe (7.5 pmol/mL) for the “house-keeping” gene glyceraldehyde 3-phosphate dehydrogenase (GAPDH-AP, 5′-CC TGC TTC ACC ACC TTC TTG ATG TCA, LGC Biosearch Technologies, Lystrup, Denmark) [49]. The specificity of the hybridization reaction was tested by applying the hybridization buffer without probe and by treatment of sections with RNase A (200 µg/mL) in RNase buffer (1:4) prior to the in situ hybridization with AP-labelled probe.

To uncover the cell architecture of the spinal cord, Nissl staining was performed on the sections adjacent to those that were used for in situ hybridization. The sections were dried at room temperature for several hours, and then immersed in 70% ethanol for 1 day. Thereafter the sections were rinsed in distilled water and stained in 1% Toluidine Blue (Sigma-Aldrich, St. Louis, MI, USA) for 3 min (pH 4.0). The sections were then dehydrated in graded ethanol and cleared in xylene and finally coverslipped with DPX (Merck Millipore, Burlington, MA, USA).

### 4.5. Data Acquisition and Analysis

All sections from immunohistochemistry and in situ hybridization were observed using a Leica DM600B light microscope and pictures were obtained using a Leica DFC420 digital microscope camera and Leica Application Suite software (version 2.8.1) (all the hardware and software were from Leica Microsystems, Wetzlar, Germany). Fluorescent images from double immunolabeled sections were taken with a Nikon A1 confocal microscope (Nikon, Tokyo, Japan).

To quantitatively analyze the abundance of the PKD2L1-IR neurons in different segments from different animal species, 3–4 animals were used from each species. PKD2L1-IR neurons from 5–20 sections from each segment (cervical, thoracic, lumbar, and sacral) of the DAB-stained sections from each animal were manually counted under the microscope. Since in cats and monkeys the PKD2L1-IR cell bodies were not always clearly labeled, we chose to count the dendritic bulbs protruding to the central canal, which were clearly and consistently labeled in all the animal species studied, to represent the numbers of the PKD2L1-IR neurons in different animal species. The average number of PKD2L1-IR neurons from each segment was calculated by averaging the numbers of the dendritic bulbs from all the sections examined and expressed as cell number/section. To quantitatively analyze the sizes of the dendritic bulbs from the different animal species, pictures of the dendritic bulbs were taken with 100× oil-immersed objective from horizontal thoracic and lumbar spinal cord sections from all the 4 animal species. In each animal species around 100 bulbs from 2–3 animals were used for the analysis. The area of a dendritic bulb was measured using ImageJ software (version 1.53t) and then its diameter was calculated according to the formula area = $\pi r^{2}$ (where *r* is the radium, diameter = 2*r*) assuming that the shape of the bulbs was spherical. The average diameter of the dendritic bulbs for each animal species was calculated from all the measured bulbs in that species. Statistical analyses were performed in GraphPad Prism (version 9.5) and a *p* value < 0.05 was considered as significant. The average number of PKD2L1-IR neurons followed Gaussian distribution and thus the significant differences among groups were investigated using ordinary one-way ANOVA with Sidak–Holm multiple comparisons. Since the diameters of the dendritic bulbs did not follow Gaussian distribution the differences among groups were investigated using the Kruskal–Wallis ranks test with Dunn’s multiple comparisons.

## Figures and Tables

**Figure 1 ijms-24-13582-f001:**
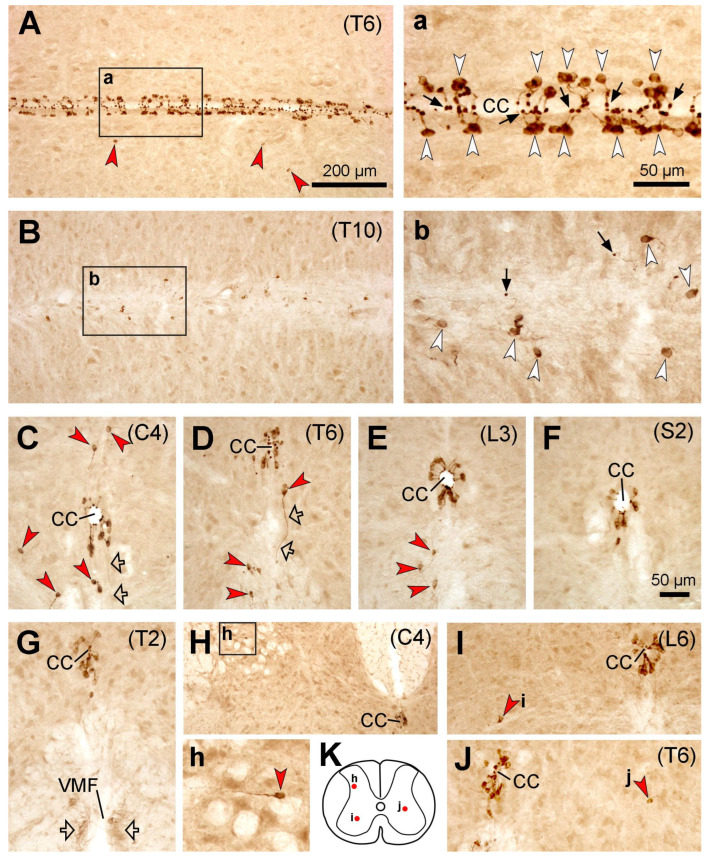
Microphotographs showing PKD2L1-immunoreactive (IR) neurons in the mouse spinal cord. (**A**,**a**) Images from a thoracic section (T6) across the central canal showing the distribution of PKD2L1-IR cell bodies (white arrowheads in (**a**)) and their bulbous protrusions (black arrows in (**a**)) in relation to the central canal (CC). (**a**) is an enlargement of the rectangular area in (**A**). Red arrowheads in (**A**) point a few PKD2L1-IR neurons away from the CC. (**B**,**b**) Images from another thoracic section (T10) ventral to the central canal showing the distribution of PKD2L1-IR cell bodies in this region. Spreading cell bodies could be seen in this region (white arrowheads in (**b**)). Occasionally small bulbs, possibly the dendritic protrusions, were also obvious (black arrows in (**b**)). (**b**) is an enlargement of the rectangular area in (**B**). (**C**–**G**) Images from transverse sections from cervical (C4 in (**C**)), thoracic (T6 in (**D**)) and T2 in (**G**)), lumbar (L3 in (**E**)), and sacral (S2 in (**F**)) levels showing PKD2L1-IR neurons around the CC and away from the CC (red arrowheads in (**C**,**D**)). Ventrally running PKD2L1-IR fibers were seen in (**C**,**D**) (hollow arrows), which could be either axons or dendrites. In (**G**), cut fiber bundles could be observed on two sides of the ventral median fissure (VMF). (**H**–**J**) A small number of PKD2L1-IR neurons were observed in the gray matter further away from the CC. These ectopic cells could be in the dorsal horn (**H**), intermediate region (**I**), or the ventral horn (**J**). (**h**) is an enlargement of the rectangular area in (**H**). (**K**) Schematic drawing depicting the relative positions of the ectopic PKD2L1-IR neurons from (**H**–**J**) in the spinal gray matter. For (**C**–**J**), the dorsal is upwards. Scale bar in (**A**), valid for (**A**,**B**,**H**), 200 µm; in (**a**), valid for (**a**,**b**,**h**), 50 µm; in (**F**), valid for (**C**–**G**,**I**,**J**), 50 µm.

**Figure 2 ijms-24-13582-f002:**
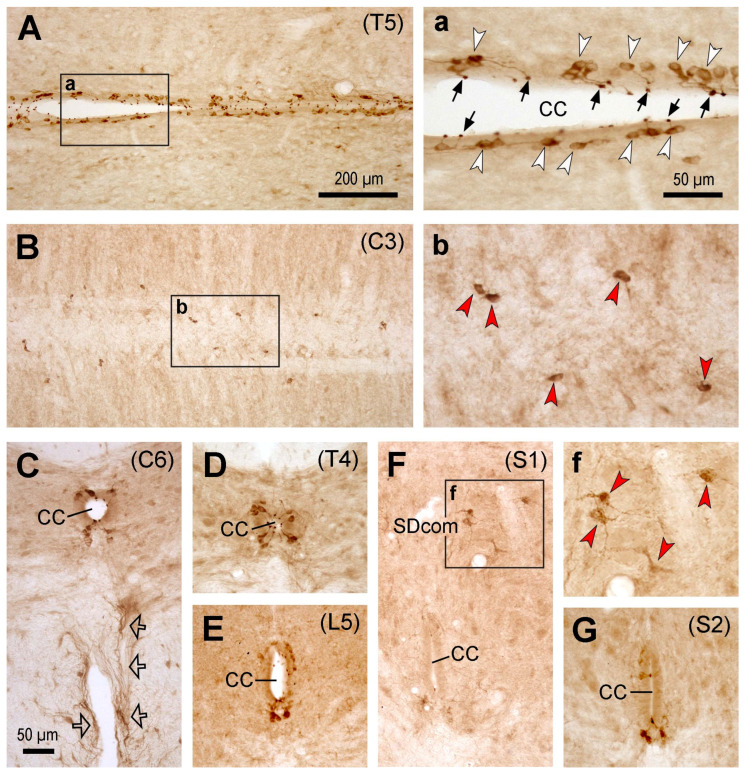
Microphotographs showing PKD2L1-immunoreactive (IR) neurons in the rat spinal cord. (**A**,**a**) Images from a thoracic section (T5) across the central canal showing the distribution of PKD2L1-IR cell bodies (white arrowheads in (**a**)) and their bulbous protrusions to the central canal (CC) (black arrows in (**a**)). (**a**) is an enlargement of the rectangular area in (**A**). (**B**,**b**) Images from a cervical section (C3) ventral to the central canal showing the distribution of PKD2L1-IR cell bodies in this region. Spreading cell bodies could be seen in this region (red arrowheads in (**b**)). (**b**) is an enlargement of the rectangular area in (**B**). (**C**–**G**) Images from transverse sections from cervical (C6 in (**C**)), thoracic (T4 in (**D**)), lumbar (L5 in (**E**)), and sacral (S1 in (**F**) and S2 in (**G**)) levels showing PKD2L1-IR neurons around the CC and away from the CC (red arrowheads in (**f**)). The ectopic PKD2L1-IR neurons were seen in the sacral dorsal commissural nucleus (SDcom) in (**F**,**f**). (**f**) is an enlargement of the rectangular area in (**F**). Ventrally running PKD2L1-IR fibers were shown in (**C**) (hollow arrows), which might be axons originating from the PKD2L1-IR neurons around the central canal. For (**C**–**G**), the dorsal is upwards. Scale bar in (**A**), valid for (**A**,**B**), 200 µm; in (**a**), valid for (**a**,**b**,**f**), 50 µm; in (**C**), valid for (**C**–**G**), 50 µm.

**Figure 3 ijms-24-13582-f003:**
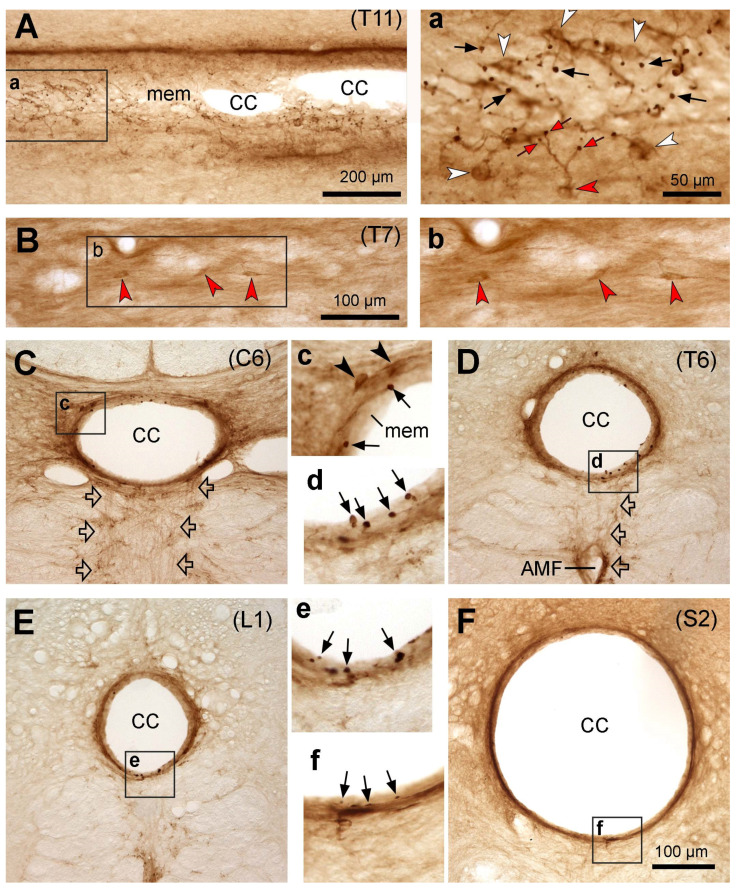
Microphotographs showing PKD2L1-immunoreactive (IR) neurons in the cat spinal cord. (**A**,**a**) Images from a horizontal thoracic section (T11) across the central canal showing the distribution of PKD2L1-IR cell bodies (white arrowheads in (**a**)) and their bulbous protrusions (black arrows in (**a**)) in relation to the central canal (CC in (**A**)). (**a**) is an enlargement of the rectangular area in (**A**). The red arrowhead in (**a**) points to a PKD2L1-IR cell body giving off branches with three bulbs (red arrows). (**B**,**b**) Images from a horizontal thoracic section (T7) showing three ectopic PKD2L1-IR neurons below the central canal (red arrowheads). (**b**) is an enlargement of the rectangular area in (**B**). (**C**–**F**) Images from transverse sections from cervical (C6 in (**C**)), thoracic (T6 in (**D**)), lumbar (L1 in (**E**)), and sacral (S2 in (**F**)) segments showing the PKD2L1-IR neurons (black arrowheads in (**c**) around the CC and the protrusion bulbs embedded in a layer of membrane in the inner surface of the central canal (black arrows in (**c**–**f**)). Thin fibers were seen running from the CC ventrally (hollow arrows in (**C**,**D**)), which might form axonal fiber bundles along the sides of the ventral median fissure (VMF). (**c**–**f**) are the enlargements of the rectangular areas in (**C**–**F**), respectively. For (**C**–**F**), the dorsal is upwards. Scale bar in (**A**), 200 µm; in (**B**), 100 µm; in (**F**), valid for (**C**–**F**), 100 µm; in (**a**), valid for (**a**–**f**), 50 µm.

**Figure 4 ijms-24-13582-f004:**
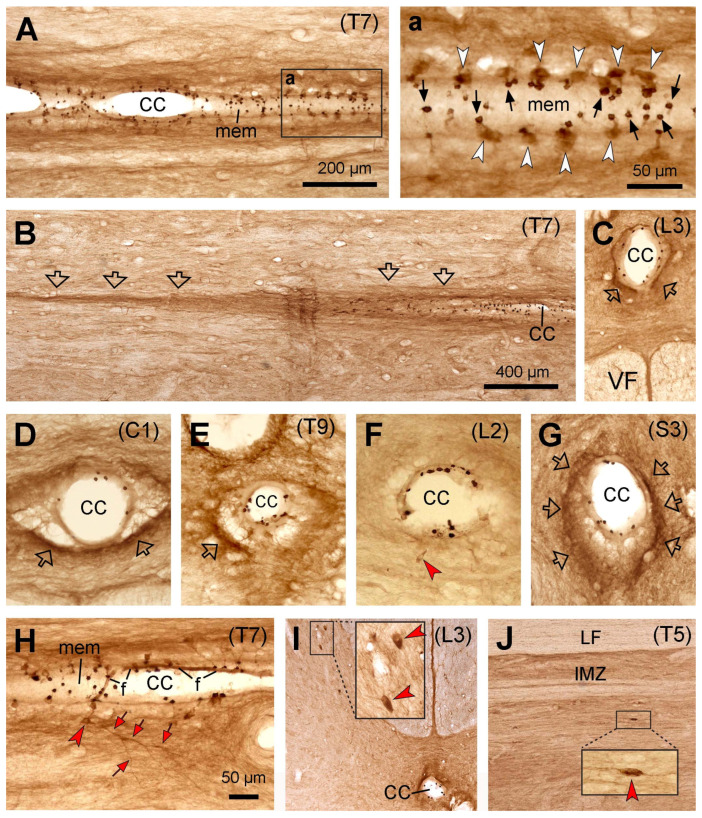
Microphotographs showing PKD2L1-immunoreactive (IR) neurons in the macaque monkey spinal cord. (**A**,**a**) Images from a thoracic horizontal section (T7) across the central canal showing the distribution of PKD2L1-IR cell bodies (white arrowheads in (**a**)) and their bulbous protrusions (black arrows in (**a**)) in relation to the central canal (CC in (**A**)). (**a**) is an enlargement of the rectangular area in (**A**). (**B**,**b**) Images from a section ventral to the section in (**A**) showing the PKD2L1-IR fibers running along the lateral and ventral sides of the CC (hollow arrows). (**C**–**E**) Images from transverse sections from cervical (C1 in (**D**)), thoracic (T9 in (**E**)), lumbar (L3 in (**C**)), L2 in (**F**)), and sacral (S3 in (**G**)) segments showing PKD2L1-IR protruding bulbs at the inside surface of the CC (dark-brown stained dots). Cut fibers were seen outside the CC on its different sides (hollow arrows in (**C**–**E**,**G**)), but no fiber bundles ran ventrally towards the ventral median fissure (**C**). VF: ventral funiculus. In (**H**) a fiber (probably an axon) with some branches originating from a PKD2L1-IR cell body could be seen running longitudinally and laterally (red arrowheads). (**I**) A few ectopic PKD2L1-IR cell bodies could be seen in the medial dorsal horn in a lumbar segment (red arrowheads in the inset). (**J**) An ectopic PKD2L1-IR cell body could be seen in the intermediate zone (IMZ) in a horizontal section from a thoracic segment (red arrowhead in the inset). LF, lateral funiculus. The PKD2L1-IR bulbs were embedded in a layer of membrane (mem in (**A**,**a**,**H**)). Occasionally a think fiber (**f**) could be seen in the CC on which many PKD2L1-IR bulbs attached (**H**). Whether this fiber is equal to a Reissner fiber needs to be confirmed further. For (**C**–**G**,**I**), the dorsal is upwards. Scale bar in (**A**), valid for (**A**,**C**), 200 µm; in (**a**), 50 µm; in (**B**), valid for (**B**,**I**,**J**), 400 µm; in (**H**), valid for (**D**–**H**) and the insets in (**I**,**J**), 50 µm.

**Figure 5 ijms-24-13582-f005:**
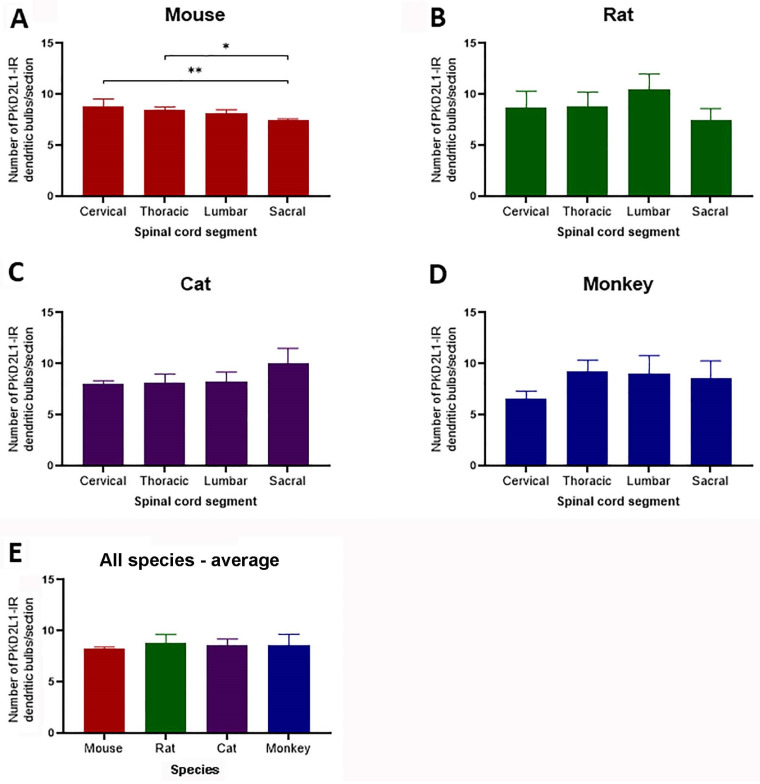
Histograms demonstrating the average number of PKD2L1-immunoreactive (IR) dendritic bulbs protruding to the central canal per 40 µm transverse section in different spinal segments in mice, rats, cats, and monkeys. (**A**) Analysis showed a significant difference between the average numbers in the spinal cord segments in mice. (**B**–**D**) No significant segmental distribution differences were found for rats, cats, and monkeys. (**E**) The number of PKD2L1-IR dendritic bulbs per transverse section averaged from the sections from the different parts of the spinal cord did not significantly differ between the species. The numbers were expressed as Mean ± SD. n = 4 for mice, n = 3 for rats, n = 3 for cats, and n = 3 for monkeys (except the cervical segment in monkey, where n = 2). Color code: red–mouse, green–rat, purple–cat, blue monkey, applies to all panels. One-way ANOVA with Sidak–Holm multiple comparison. * *p* < 0.05, ** *p* < 0.01.

**Figure 6 ijms-24-13582-f006:**
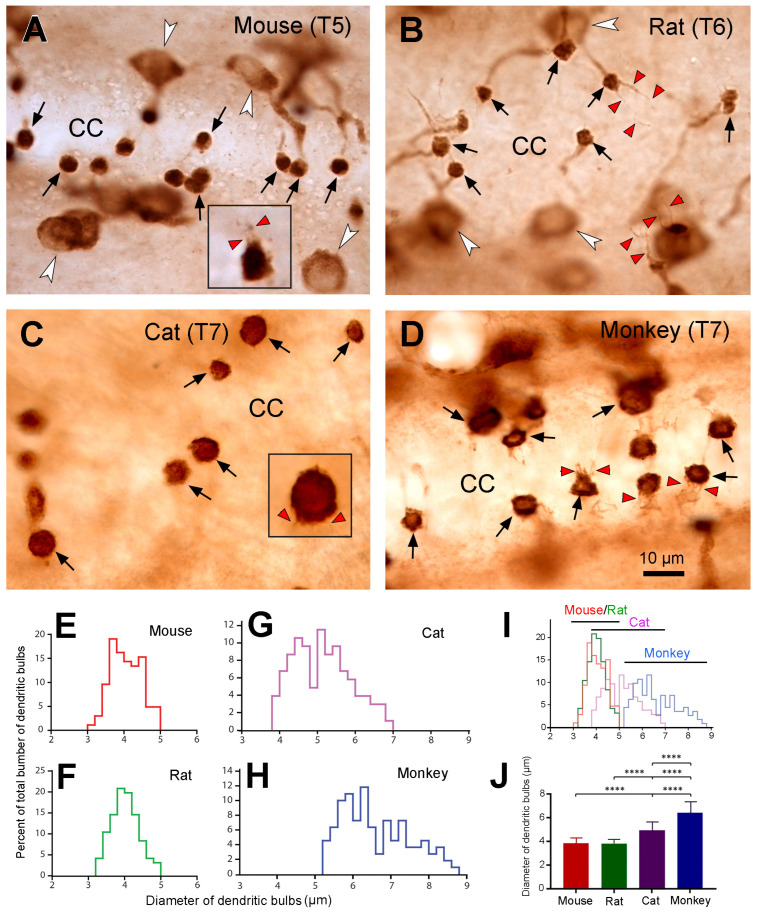
Morphology and sizes of PKD2L1-immunoreactive (IR) dendritic bulbs in the central canal in different animal species. (**A**–**D**) High-magnification microphotographs showing PKD2L1-IR dendritic bulbs protruded to the spinal cord central canal from thoracic horizontal sections from a mouse, a rat, a cat, and a macaque monkey. The shape of the bulbs (black arrows) was mostly round. In the mouse and rat, the sizes of the bulbs were smaller, whereas in the cat and monkey they were relatively larger. In all the species cilium-like thin fibers were seen originating from the bulbs although the lengths of these fibers were not the same. For example, in the rat (**B**) the fibers were longer than in the other species (red arrowheads); and in some cases, two long fibers could be seen originating from one bulb (red arrowheads in the upper right and lower right corner in (**B**)). In the cat shorter fibers forming crown-like radiations from the surface of the bulb ((**C**), red arrowhead in inset), and in the monkey medium-long fibers from the bulbs forming a tuft from the surface of the bulbs (red arrowheads in (**D**)). Occasionally such a tuft structure was also seen in the bulbs in mice (red arrowheads in the inset in (**A**)). White arrowheads in (**A**,**B**) point to PKD2L1-IR cell bodies. CC: central canal or close to its border. Insets in (**A**,**C**) were from different sections in a mouse and cat showing the shape of the cilium-like fibers from the surface of the dendritic bulbs with a double amplification as in (**A**–**D**). Scale bar in (**D**), valid for (**A**–**D**), 10 µm. (**E**–**I**) Percent distribution of dendritic bulbs as a function of the diameters of the bulbs in mice, rats, cats, and monkeys. The total number of the dendritic bulbs calculated was 106, 96, 105, and 110 in mice, rats, cats, and monkeys, respectively. It could be seen that the size of the dendritic bulbs was smaller in mice and rats (3–5 µm), medium in cats (3.8–7 µm) and larger in monkeys (5.2–8.8 µm). From panel (**I**) it could be seen that there was not an overlap zone for the diameters of the bulbs from mice/rats and monkeys. (**J**) Graph showing the sizes of the dendritic bulbs in the 4 animal species. Color code: red–mouse, green–rat, purple–cat, blue–monkey, applies to all panels. Mean ± S.D. Kruskal-Wallis ranks test with Dunn’s multiple comparison indicated that there was a statistically significant difference in the diameters of the dendritic bulbs among mice, rats, cats, and monkeys. The size of dendritic bulbs was significantly larger in monkeys compared to all the three other animal species and the size was significantly larger in cats compared to mice and rats. No difference was found between mice and rats. **** *p* < 0.0001.

**Figure 7 ijms-24-13582-f007:**
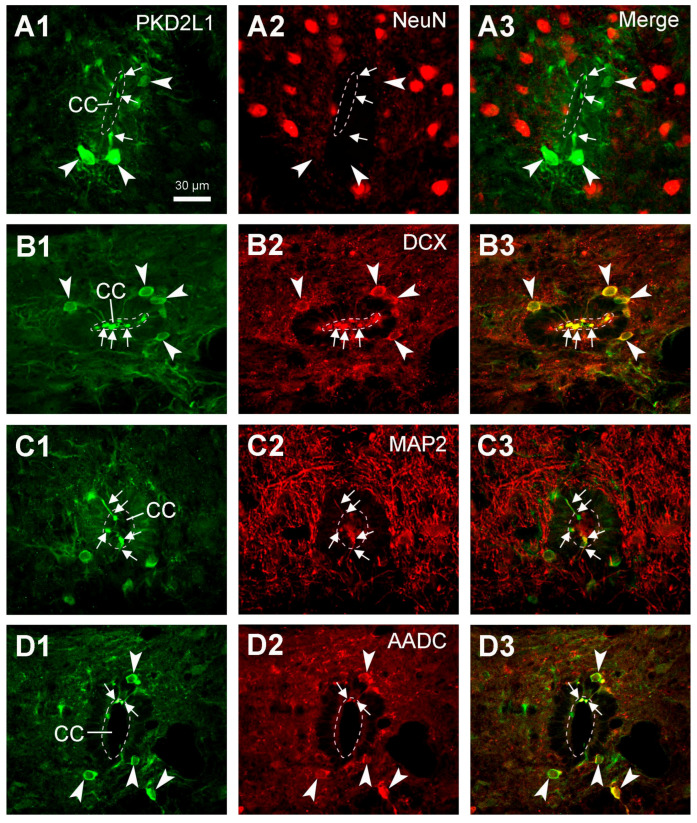
Confocal images of double-immunostaining showing the properties of PKD2L1-immunoreactive (IR) neurons in the rat spinal cord. (**A1**–**A3**) In a thoracic section, PKD2L1-IR neurons were not double labeled with NeuN antibody. (**B1**–**B3**) In a thoracic section, PKD2L1-IR neurons were double labeled with DCX antibody, indicating their immature nature. (**C1**–**C3**) In a lumbar section, the PKD2L1-IR protrusions, including the bulbous endings in the central canal were double labeled with MAP2 antibody. (**D1**–**D3**) In a thoracic section, PKD2L1-IR neurons, including their protrusions to the central canal, were double labeled with AADC antibody. For all panels, the large white arrowheads indicate the locations of cell bodies, and the small white arrows indicate the protrusions including the bulbous endings. Scale bar in (**A1**), valid for all panels, 30 µm.

**Figure 8 ijms-24-13582-f008:**
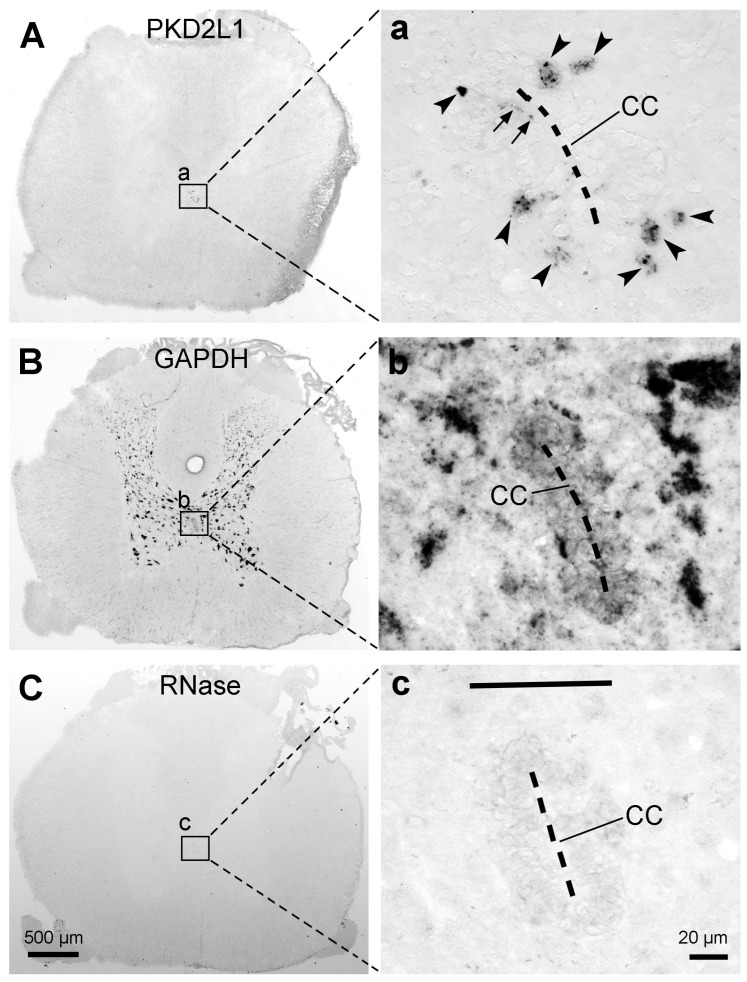
Expression of PKD2L1 mRNA with in situ hybridization in rats. (**A**,**a**) Images showing PKD2L1 mRNA expression in a thoracic section. (**a**) High magnification of the framed area in (**A**) showing PKD2L1 mRNA expression around the central canal (CC, dashed line). Arrowheads indicate PKD2L1 mRNA-expressing cell bodies, and the arrows indicate a presumable PKD2L1 mRNA-expressing dendritic protrusion towards the CC. (**B**,**b**) Images from a thoracic section hybridized for GAPDH mRNA as a procedural control. Most of the cell bodies in the gray matter express GAPDH mRNA. (**C**,**c**) Control showing absence of hybridization signal around the CC in a thoracic section treated with RNase A and then hybridized for PKD2L1 mRNA. Scale bar in (**C**), valid for (**A**–**C**), 500 µm; in (**c**), valid for (**a**–**c**), 20 µm.

## Data Availability

The data presented in this study are available on request from the corresponding author.

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
