# Peer review of "A Comparison of PKD2L1-Expressing Cerebrospinal Fluid Contacting Neurons in Spinal Cords of Rodents, Carnivores, and Primates"

_ijms, 2023, doi:10.3390/ijms241713582_

Round 1
Reviewer 1 Report
In this manuscript, the author investigated the distribution of polycystic kidney disease 2-like 1 immunoreactive (PKD2L1-IR) cerebrospinal fluid contacting neurons (CSF-cNs) in the spinal cord of four different mammalian animal species, which included mouse, rat, cat, and macaque monkey.
They found, that that PKD2L1-expressing CSF-cNs were present at all levels of the spinal cord in these animal species. PKD2L1-IR neurons were found in the rostrocaudal extent of the spinal cord in the examined species, where they were predominantly located around the central canal, although different. The relatively larger size of dendritic spines in cats and macaques compared to mice and rats indicates that individual PKD2L1 neurons receive input stimulation with greater capacity in larger animals. In addition, rat in situ hybridization revealed PKD2L1 mRNA expression in cells around the central canal.
These results indicate that intrinsic sensory neurons are conserved in non-mammalian and mammalian vertebrates and may play a fundamental role in the regulation of motor and sensory functions and the maintenance of central nervous system homeostasis.
The topic is timely and important and may attract much attention.The article is clearly and understandably written, and illustrated and has no serious comments.
I only have some question:
1. To what extent previous electrophysiological experiments in cats and macaques could have influenced PKD2L1 expression?
2. There is a big difference in animal weight between cats and rats. Could this have affected the number of CSF-cNs containing PKD2L1?
3. It would be worth giving the age for mice, rats and cats as well.
4. In the case of cats, it is not entirely clear based on the description whether the animals were perfused immediately after the electrophysiological examination in 2008-2009, or much later? Have there been any experiments with cats so far?
5. Although it is only about tissue processing,, in the case of cats and macaques, what about the animal experiment permit? Did the two other institutes remove the spinal cord and did they have permission to do so, or did the authors have permission to experiment with these animals and process their samples?
Minor editing of English language required. There are some grammatical and wording errors.
Author Response
Response to reviewers’ comments
Reviewer # 1:
In this manuscript, the author investigated the distribution of polycystic kidney disease 2-like 1 immunoreactive (PKD2L1-IR) cerebrospinal fluid contacting neurons (CSF-cNs) in the spinal cord of four different mammalian animal species, which included mouse, rat, cat, and macaque monkey.
They found that PKD2L1-expressing CSF-cNs were present at all levels of the spinal cord in these animal species. PKD2L1-IR neurons were found in the rostrocaudal extent of the spinal cord in the examined species, where they were predominantly located around the central canal, although different. The relatively larger size of dendritic spines in cats and macaques compared to mice and rats indicates that individual PKD2L1 neurons receive input stimulation with greater capacity in larger animals. In addition, rat in situ hybridization revealed PKD2L1 mRNA expression in cells around the central canal.
These results indicate that intrinsic sensory neurons are conserved in non-mammalian and mammalian vertebrates and may play a fundamental role in the regulation of motor and sensory functions and the maintenance of central nervous system homeostasis.
The topic is timely and important and may attract much attention. The article is clearly and understandably written, and illustrated and has no serious comments.
Response: Thanks for the nice comments. We really appreciate them.
I only have some question:
- To what extent previous electrophysiological experiments in cats and macaques could have influenced PKD2L1 expression?
Response: This is a very good question and may need to be addressed in the future study. To frankly answer this question, we really don’t know to what extent previous electrophysiological experiments in cats and macaques could have influenced PKD2L1 expression. To address this issue, we need to have naïve cats and monkeys as control animals, which we cannot have now. This statement is added in the Discussion (pages 19-20, lines 486-492).
- There is a big difference in animal weight between cats and rats. Could this have affected the number of CSF-cNs containing PKD2L1?
Response: We believe that this issue was already addressed in the Discussion. See page 20, lines 508-515, page 21, lines 574-583.
- It would be worth giving the age for mice, rats and cats as well.
Response: It is done (see page 24)
- In the case of cats, it is not entirely clear based on the description whether the animals were perfused immediately after the electrophysiological examination in 2008-2009, or much later? Have there been any experiments with cats so far?
Response: This is added in page 24, line 714. There were no cat experiments since 2009 due to the ethical issue in Denmark. Thus, we cannot get fresh tissue samples from cats.
- Although it is only about tissue processing, in the case of cats and macaques, what about the animal experiment permit? Did the two other institutes remove the spinal cord and did they have permission to do so, or did the authors have permission to experiment with these animals and process their samples?
Response: The two other institutes removed the spinal cord after perfusing the animals, and they had the permission to do so. Now the permission numbers were added in the related parts in Materials and Methods (page 24, line712-713, 721-724). At the University of Southern Denmark, we don’t need a permission to process tissue samples from dead animals.
Minor editing of English language required. There are some grammatical and wording errors.
Response: The grammatical errors were carefully checked and corrected throughout the manuscript.
Reviewer 2 Report
This article showcases an interesting study on a topic that has garnered substantial attention over recent decades. While prior research primarily focused on mice, the authors have notably broadened the scope by delving into other mammalian models, particularly cats and macaques. This research systematically incorporates techniques such as immunohistochemistry, immunofluorescence, and in situ hybridization. Furthermore, it offers compelling morphometric data.
The writing style is clear and accessible, avoiding unnecessary verbosity. The featured images are of high quality, clearly marked, and aptly identified. The bibliography stands out as both current and comprehensive.
Abstract: The abstract is exhaustive in terms of methodology and findings. It outlines the techniques used (immunohistochemistry and in-situ hybridization) and clearly specifies the species studied. However, there is some room for improvement:
- While differences between species in terms of PKD2L1 distribution and expression are noted, would it not be valuable to provide more insight into these differences and their potential implications?
- The abstract concludes by stating that intrinsic sensory neurons might play an essential role in various functions and homeostasis. Nonetheless, it would be beneficial to offer some information about how the findings support this assertion.
Introduction:
The introduction provides a detailed and comprehensive overview of the cerebrospinal fluid contacting neurons (CSF-cNs) and clearly states the main purpose of the study: to map the distribution of PKD2L1-expressing CSF-cNs in the spinal cord of four species. The authors put the role of the CSF-cNs in context, from their location to their biochemical properties and sensory functions.
However, the introduction delves deeply into the PKD2L1 channel, its functions, and its expression across different species. While this channel is clearly central to the study, the authors use in their study other markers, whose relevance and role in the CSF-cNs should be minimally introduced.
Results
The results are presented in a systematic manner, well-illustrated, and impeccably labeled. The only query I have pertains to the expression of DCX in rats (Fig. 7B2). Isn't it too pronounced for an adult? Shouldn't this have been addressed by the authors?
Why was confocal immunostaining only performed on rat sections? Given the conditions of part of the samples -ultra-frozen specimens from cats and macaques-, the application of confocal immunolabelling would seem a probable choice. It offers potentially greater sensitivity and specificity in scenarios where light immunohistochemistry might falter due to sample degradation. Should the authors not have performed confocal immunostaining for both species? It's probable that they did. If so, what were the results? If not, why did they refrain from trying it? Such information would be valuable for discussions in future studies
Discussion
The discussion is thorough, detailed, and comprehensive, building upon existing knowledge while highlighting the novelty and implications of the current findings. The careful comparative analysis with previous studies and acknowledgment of limitations is commendable. Personally, I find this type of discussion highly commendable. It possesses a level of thoroughness that demonstrates a deep understanding of the subject matter. Not only is it systematic in its approach, but it also possesses a self-critical approach. This is invaluable as ensures the content's robustness and reliability.
I concur with the authors that the most intriguing and noteworthy discovery in the study was the variation in dendritic bulb sizes across different animal species. However, it's essential to acknowledge that the processes of freezing, cryoprotection, and subsequent thawing for the cat and macaque histological samples might have influenced the tissue's volume and morphology. This factor should be explicitly addressed when evaluating the morphometry results in this critical section.
Material and Methods
In the materials and methods section, everything is detailed and rigorously explained.
A problematic issue arises concerning the thickness of the sections. In the case of mice, these were done at 20 microns, while for the rest of the species it was 40 microns. This discrepancy presents a limitation when conducting the morphometric study, as logically, the section thickness will bias the measurements obtained. That should be acknowledged.
Finally, the authors do not describe any type of control for the histological techniques, neither positive nor negative. Were these controls performed? If not, this should be indicated.
Minor issues:
The email address for Mengliang Zhang in the correspondence section contains an error. It's listed as mzhang@heaalth.sdu.dk. Instead of "heaalth" it should be "health".
Author Response
Response to reviewers’ comments
Reviewer #2
This article showcases an interesting study on a topic that has garnered substantial attention over recent decades. While prior research primarily focused on mice, the authors have notably broadened the scope by delving into other mammalian models, particularly cats and macaques. This research systematically incorporates techniques such as immunohistochemistry, immunofluorescence, and in situ hybridization. Furthermore, it offers compelling morphometric data.
The writing style is clear and accessible, avoiding unnecessary verbosity. The featured images are of high quality, clearly marked, and aptly identified. The bibliography stands out as both current and comprehensive.
Response: Thanks for the very nice comments. We appreciate them very much.
Abstract:
The abstract is exhaustive in terms of methodology and findings. It outlines the techniques used (immunohistochemistry and in-situ hybridization) and clearly specifies the species studied. However, there is some room for improvement:
- While differences between species in terms of PKD2L1 distribution and expression are noted, would it not be valuable to provide more insight into these differences and their potential implications?
- The abstract concludes by stating that intrinsic sensory neurons might play an essential role in various functions and homeostasis. Nonetheless, it would be beneficial to offer some information about how the findings support this assertion.
Response: We agree with these suggestions. Accordingly, we have added two sentences in the abstract to address these issues. We don’t know whether these successfully addressed the reviewer’s comments but that is what we interpreted to be included.
Introduction:
The introduction provides a detailed and comprehensive overview of the cerebrospinal fluid contacting neurons (CSF-cNs) and clearly states the main purpose of the study: to map the distribution of PKD2L1-expressing CSF-cNs in the spinal cord of four species. The authors put the role of the CSF-cNs in context, from their location to their biochemical properties and sensory functions.
However, the introduction delves deeply into the PKD2L1 channel, its functions, and its expression across different species. While this channel is clearly central to the study, the authors use in their study other markers, whose relevance and role in the CSF-cNs should be minimally introduced.
Response: The relevance and role of other markers (NeuN, MaP2, DCX, and AADC) than PKD2L1 in the CSF-cNs were briefly introduced in the Introduction (page 3, line 102-106).
Results:
The results are presented in a systematic manner, well-illustrated, and impeccably labeled. The only query I have pertains to the expression of DCX in rats (Fig. 7B2). Isn't it too pronounced for an adult? Shouldn't this have been addressed by the authors?
Response: This issue has been addressed in the Discussion (page 22, lines 628-633).
Why was confocal immunostaining only performed on rat sections? Given the conditions of part of the samples -ultra-frozen specimens from cats and macaques-, the application of confocal immunolabelling would seem a probable choice. It offers potentially greater sensitivity and specificity in scenarios where light immunohistochemistry might falter due to sample degradation. Should the authors not have performed confocal immunostaining for both species? It's probable that they did. If so, what were the results? If not, why did they refrain from trying it? Such information would be valuable for discussions in future studies.
Response: We fully agree with the reviewer opinion. We have tried double-fluorescent immunohistochemistry in cat and macaque spinal tissues in the beginning. However, due to that in most cases the cell bodies of PKD2L1-IR neurons were poorly labeled we decided not to add the related data in this manuscript, because only showing the dendritic bulbs makes not much sense. Another reason is that the manuscript is already very long and comprehensive. This part of work will be for sure in our future research plan.
Discussion:
The discussion is thorough, detailed, and comprehensive, building upon existing knowledge while highlighting the novelty and implications of the current findings. The careful comparative analysis with previous studies and acknowledgment of limitations is commendable. Personally, I find this type of discussion highly commendable. It possesses a level of thoroughness that demonstrates a deep understanding of the subject matter. Not only is it systematic in its approach, but it also possesses a self-critical approach. This is invaluable as ensures the content's robustness and reliability.
I concur with the authors that the most intriguing and noteworthy discovery in the study was the variation in dendritic bulb sizes across different animal species. However, it's essential to acknowledge that the processes of freezing, cryoprotection, and subsequent thawing for the cat and macaque histological samples might have influenced the tissue's volume and morphology. This factor should be explicitly addressed when evaluating the morphometry results in this critical section.
Response: Very good comments. We have added some sentences in the Discussion to address this issue (page 21, lines 586-594).
Material and Methods:
In the materials and methods section, everything is detailed and rigorously explained.
A problematic issue arises concerning the thickness of the sections. In the case of mice, these were done at 20 microns, while for the rest of the species it was 40 microns. This discrepancy presents a limitation when conducting the morphometric study, as logically, the section thickness will bias the measurements obtained. That should be acknowledged.
Response: We believe that the reviewer mixed the sections from mice and rats. The sections from mouse spinal cords were always cut into 40 µm for immunohistochemistry. However, for rat spinal cords, we cut 40 µm for immunohistochemistry and 20 µm for in-situ hybridization. In page 24, lines 705-706, we wrote: “The rats used for immunohistochemistry were perfused with 4% PFA, and the spinal cords were processed the same way as the mice”. Maybe this sentence is not very straight forward.
Finally, the authors do not describe any type of control for the histological techniques, neither positive nor negative. Were these controls performed? If not, this should be indicated.
Response: We performed negative controls for all the immunohistochemistry as a lab routine, but not positive controls especially for the commonly used primary antibodies. This is indicated in page 25, lines 753-756.
Minor issues:
The email address for Mengliang Zhang in the correspondence section contains an error. It's listed as mzhang@heaalth.sdu.dk. Instead of "heaalth" it should be "health".
Response: It is corrected.